# Beneficial Effects of Enoki Mushroom Extract on Male Menopausal Symptoms in Japanese Subjects: A Randomized, Double-Blind, Placebo-Controlled Study

**DOI:** 10.3390/nu17071208

**Published:** 2025-03-30

**Authors:** Shizuo Yamada, Michiyo Shirai, Koji Nagashima, Jun Mochizuki, Ken Ono, Shinji Kageyama

**Affiliations:** 1Center for Pharma-Food Research, Graduate School of Pharmaceutical Sciences, University of Shizuoka, Shizuoka 422-8526, Japan; cpfr02@u-shizuoka-ken.ac.jp; 2TechnoSuruga Laboratory Co., Ltd., Shizuoka 424-0065, Japan; koji.nagashima@tecsrg.co.jp (K.N.); jm_1001@tecsrg.co.jp (J.M.); 3Izu Health & Medical Center, Shizuoka 410-0638, Japan; kenken@izu-hmc.ecnet.jp; 4Kageyama Urology Clinic, Shizuoka 420-0838, Japan; kage3309@vmail.plala.or.jp

**Keywords:** enoki mushroom, menopause, AMS, testosterone

## Abstract

With the increase in average life expectancy, age-related male menopause has become a cause of decreased quality of life in men. The present study investigated the efficacy and safety of powdered enoki mushroom extract containing adenosine (test food) for menopausal symptoms in middle-aged and elderly men based on an evaluation of Heinemann’s Aging Males’ Symptoms (AMS) scores. The test food and placebo were administered to healthy men with AMS scores of 27–49 for 12 weeks. AMS score (primary endpoint) and testosterone level (secondary endpoint) were evaluated before and 12 weeks after the intake of the test food and placebo. The intake of the test food for 12 weeks significantly improved the sexual subscale of the AMS. In the cumulative χ2 test, the number of subjects showing high improvement was significantly higher in the test food group than in the placebo food group. In a stratified analysis of subjects divided into two groups based on a change in total testosterone levels of <0.5 ng/mL and ≥0.5 ng/mL after the intake of the test food, the number of subjects with increased total testosterone levels of ≥0.5 ng/mL was significantly higher in the test food group than in the placebo group. These results suggest the beneficial effects of enoki mushroom extract on symptoms of male menopause.

## 1. Introduction

With the increase in average life expectancy, age-related male menopause has become a cause of decreased quality of life in men. Male menopause is characterized by depression, psychological symptoms, muscle weakness, sleep disorders, erectile dysfunction, and other menopausal symptoms, as well as the psychological burden and decline in male hormones that come with aging [1]. Late-onset hypogonadism syndrome (LOH) is a condition of multiple organ dysfunction that is characterized by low testosterone levels and is associated with general fatigue, insomnia, depressive symptoms, cognitive decline, and lifestyle-related diseases, which collectively result in male frailty [2,3]. Androgens play important physiological functions in the body, such as muscle, bone, skin, and sexual function, and are biosynthesized from cholesterol. Among them, the decline of testosterone may be related to sexual dysfunction as well as a loss of muscle mass, cognitive decline, and other physiological dysfunctions. Previous studies have demonstrated that age-related decreases in testosterone levels induce metabolic syndrome, contribute to the development of dementia, and lead to obesity due to decreased insulin sensitivity [4,5,6]. Diet and exercise are crucial for delaying and preventing the age-related decline in testosterone levels. Generally, protein, with its natural polyphenolic antioxidants, is considered a key nutrient for the aging population. Therefore, the development of novel naturally derived functional foods and ingredients with sufficient evidence from the perspectives of safety and functionality is important.

The enoki mushroom (*Flammulina velutipes*), one of the main edible mushrooms on the planet, has long been recognized for its nutritional value and delicious taste [7]. Previous studies on enoki mushrooms have shown that it possesses various biological and pharmacological properties, including anticancer, antimicrobial, antioxidant, and immunomodulatory activities [7,8,9,10]. Iguchi et al. [11] recently reported that the administration of an ethanolic extract of enoki mushroom increased the production of testosterone in a cisplatin-impaired model of mice and suggested the involvement of adenosine as the most significant active component. The administration of enoki mushroom extract or adenosine itself to wet-floor-fatigue-model mice was also shown to promote testicular testosterone production and enhance Leydig cell function. Based on these findings, the authors concluded that enoki mushrooms, containing a high content of adenosine, may be useful against aging and fatigue. Therefore, the present study investigated the efficacy and safety of enoki mushroom extract containing adenosine (test food) for menopausal symptoms in middle-aged and elderly men based on an evaluation of Heinemann’s Aging Males’ Symptoms (AMS) scores [12,13].

## 2. Materials and Methods

### 2.1. Test Food

The test food used in this study was a powdered enoki mushroom extract (125 mg per capsule), which contained 0.56 mg of adenosine as the main pharmacologically active ingredient. The test food and placebo (including 282 mg of resistant dextrin per capsule) were administered as 10 capsules each once a day for 12 weeks between 13 June 2023 and 23 October 2023. The daily oral dose (1250 mg) of enoki mushroom extract was determined by converting the per body weight of humans and rats on the basis of the adenosine content in a previous preclinical study [11]. These products were kindly provided by TechnoSuruga Laboratory, Co., Ltd. (Shizuoka, Japan).

Enoki mushroom is consumed as a food product, and the safety of food containing enoki mushroom extract was confirmed in a single oral administration toxicity study on rats, a 9-day repeated-administration toxicity study, a genotoxicity study on mice, and a safety study in humans involving the overconsumption of foods containing enoki mushroom extracts [14]. The safety of tea beverages containing enoki mushroom extract (chitoglucan) was confirmed in a safety study on overdose in healthy subjects [15].

### 2.2. Study Design

A randomized, double-blind, placebo-controlled, parallel-group comparison study was conducted. The protocol was approved by the Clinical Research Review Committee of the Hamamatsu University School of Medicine on 23 January 2023 as special clinical research, and was conducted after the Japan Registry of Clinical Trials (jRCTs041220142) was published on 14 February 2023. In accordance with the Declaration of Helsinki and the “Clinical Research Act and the Enforcement Regulations of the Clinical Research Act”, the present study was performed with due consideration for the human rights, safety, and welfare of the subjects. The subjects were recruited at clinical institutions and local organizations, and written consent was obtained from those who desired to participate after receiving a full explanation of the study content.

### 2.3. Subjects

The target number of subjects was the mean and standard deviation (29.0 ± 5.39 and 32.4 ± 5.88) obtained from the literature for the primary endpoint and the number of cases [16]. Therefore, the target number was 56 subjects, taking into account discontinuations and dropouts. Pharmacological and basic statistical analyses were used. The primary selection criteria were healthy men (≥40 years) with AMS scores of 27–49 who had mild or moderate mesopausal symptoms (definition of late-onset hypogonadism by “LOH Syndrome Clinical Practice Guidelines Working Committee”, https://www.urol.or.jp/lib/files/other/guideline/30_loh_syndrome.pdf (in Japanese, accessed on 23 February 2025) [17]. After obtaining consent forms at the time of their recruitment, subjects who met the selection criteria and had none of the exclusion criteria were selected.

### 2.4. Selection Criteria

*Written informed consent obtained from each individual before participating in the study.

*Age ≥ 40 years with awareness of menopausal symptoms.

*Individuals able to consume the test food during the study period.

*Ability to write a diary during the intake period.

*AMS score of 27–49.

Menopausal symptoms include sexual dysfunction (erectile dysfunction and a decreased libido), mental symptoms (such as decreased motivation, decreased concentration, depressed mood, and insomnia), and physical symptoms (including sweating, dizziness, palpitations, stiff shoulders, and muscle weakness).

### 2.5. Exclusion Criteria

*Individuals undergoing treatment for male menopause.

*Individuals within 2 months since the completion of treatment for male menopause.

*Individuals who are visiting a hospital for a lifestyle-related disease and have a lifestyle-related disease.

*Individuals taking medicines, newly designated quasi-drugs, herbal medicines, health foods, or supplements related to the promotion of testosterone secretion.

*Individuals who are participating or will participate in another clinical study at the time of the initiation of this study.

*Patients with prostate-specific antigen (PSA) levels ≥ 4.0 ng/mL.

*Individuals deemed as being inappropriate by the principal investigator.

Lifestyle-related diseases are a group of diseases in which lifestyle habits, such as eating habits, exercise habits, rest, smoking, and alcohol consumption, contribute to their onset and progression (including diabetes, obesity, dyslipidemia, hypertension, myocardial infarction, stroke, lung cancer, chronic bronchitis, colon cancer, liver cirrhosis, and fatty liver).

### 2.6. Randomization and Blinding

Blinding and randomization were conducted by the researcher responsible for allocation under the clinical research protocol. Fifty-six healthy subjects who were suitable according to the selection criteria were randomly assigned to the test food and placebo groups using the substitution block method. The assignment list was sealed until the end of the study and was opened after the study was ended. Significant differences were not observed in age or sex between the subjects in each group.

### 2.7. Evaluation Method

Evaluations were performed before and after 12 weeks of intake. Compliance was confirmed by telephone at the midpoint of the study. The study was terminated if the subject withdrew consent or stopped consuming the test food. Complications or adverse events by the subjects were recorded. If the subjects were taking any medications for other diseases before the start of the study, their dose and administration were unchanged throughout the study period.

AMS [12,13] as the primary endpoint consists of 17 items, namely psychological (5 items), physical (7 items), and sexual function (5 items) factors, and assesses the severity of male menopause (LOH symptoms). The scoring scheme for the AMS scale is as follows: The questionnaire rates each of the 17 items on a severity scale of 1 to 5. The composite scores for each of the three dimensions (subscales) are obtained by adding the scores of the items of the respective dimensions. The composite score (total score) is the sum of the three dimension scores, namely, the psychological, somatic, and sexual subscales. Secondary assessment tests comprised total and free testosterone levels (total and free testosterone). As secondary endpoints, we measured the body mass index (BMI), Beck Depression Inventory (BDI-II), Nocturia Quality-of-Life Questionnaire (N-QOL), Pittsburgh Sleep Quality Index (PSQI), and Activity Scale for the elderly (ASE).

The general examination consisted of height, weight, body mass index (BMI), blood pressure, and pulse measurements. Blood tests included white blood cell count, red blood cell count, and platelet count, as well as measurements of hemoglobin (Hb), hematocrit (Ht), uric acid, urea nitrogen, aspartate aminotransferase (AST), alanine aminotransferase (ALT), gamma-glutamyl transpeptidase (g-GTP), alkaline phosphatase (ALP), lactate dehydrogenase (LDH), total bilirubin, total protein, albumin, estimated glomerular filtration rate (eGFR), creatine phosphokinase (CPK), total-cholesterol, high density lipoprotein (HDL) cholesterol, low density lipoprotein (LDL) cholesterol, triglycerides, blood glucose, hemoglobin (Hb)A1c, amylase, Na, Cl, K, Mg, Ca, Fe, total testosterone, and free testosterone. Urinalysis included qualitative assessments of protein, sugar, urobilinogen, bilirubin, occult blood reaction, and ketone bodies, as well as specific gravity and pH. General, blood, urinalysis, and cognitive function tests were performed twice, before and after 12 weeks of intake. The safety endpoints were adverse events, adverse reactions, serious adverse events, and the number and nature of serious adverse reactions, which were evaluated by general and blood tests.

### 2.8. Statistical Analysis

To compare AMS scores between the test food and placebo groups, we conducted an analysis of covariance (ANCOVA) after the end of food intake as a covariate of the AMS scores of the subjects before food intake. The cumulative chi-squared (χ^2^) test is a valid hypothetical test method for the analysis of ordinal data in pharmacological studies [18,19,20]. We used this method to examine the incidence of subjects corresponding to each degree by ranking the changes in the scores for AMS questions before and after 12 weeks of intake in the test food and placebo groups. An exploratory stratification analysis was also performed using the mean change in total testosterone in subjects with low testosterone levels (the borderline LOH syndrome group: 8.5 pg/mL ≤ free testosterone < 11.8 pg/mL). We performed a chi-squared test on the number of cases showing a mean change in total testosterone of 0.5 ng/mL in the test food and placebo groups. The statistical analysis software used was SPSS Statistics, ver. 25, and Pharmaco Basic, ver. 17. All tests were two-tailed with a significance level < 5%.

## 3. Results

### 3.1. Subjects

Figure 1 shows the flow diagram of the subjects. This study was conducted as designed, with no changes to the protocol. There were 56 subjects, excluding 1 who withdrew his consent before the intake protocol. Therefore, the total number of subjects in the full analysis set, per protocol set (PPS), and safety analysis set was 55. There were 28 subjects in the test food group and 27 in the placebo group, with mean ages of 62.0 and 61.5 years, respectively.

Thirteen subjects were 40–59 years, twelve were 60–74 years, and three were ≥75 years old in the test food group, while twelve were 40–59 years, twelve were 60–74 years, and three were ≥75 years old in the placebo group. An analysis of baseline characteristics showed no significant differences between the test food and placebo groups. Beck Depression Inventory values were significantly higher in the test food group than in the placebo group (*p* = 0.04) (Table 1).

### 3.2. Effectiveness Evaluation

Differences between the test food and placebo groups were analyzed via ANCOVA using the “previous observation value” as the covariate for the primary endpoint, the AMS score. After confirming the linearity of the regression lines, the significance of the total AMS score and changes in the AMS subscales (physical, psychological, and sexual function factors) were examined. The results obtained did not confirm the significance of the test food in the target population PPS versus the placebo (Table 2). The mean values for the sexual function factors of AMS in the borderline LOH syndrome group (8.5 pg/mL ≤ free testosterone < 11.8 pg/mL) were significantly higher in the test food group than in the placebo group (*p* = 0.046) (Table 3).

Table 4 shows that the number of subjects with improvements of three or four points was larger in the test food group than in the placebo group. As shown in Table 5, the cumulative χ^2^ test showed that the number of subjects who responded with an improvement of four or five points was significantly higher in the test food group than in the placebo group (χ^2^_adj_ = 11.35, df = 3.68, *p* = 0.018).

There were no significant differences between the test food and placebo groups in either their total or free testosterone levels (Table 6). On the other hand, a comparison of the number of subjects separated by a mean change of 0.5 ng/mL in total testosterone showed a significantly higher number of subjects with increased testosterone levels of ≥0.5 ng/mL in the test food group than in the placebo group (χ^2^ = 4.134, df = 1, *p* = 0.042) (Table 7).

### 3.3. Safety Evaluation

A general examination of blood and urine before and after 12 weeks of intake showed no significant differences between the test food and placebo groups. Additionally, few adverse events were observed in both groups.

## 4. Discussion

The main result of the present study was that the intake of enoki mushroom extract for 12 weeks by healthy male subjects with AMS scores of 27–49 attenuated male menopausal symptoms, such as declines in sexual function and testosterone levels with aging.

Testosterone production decreases with aging [1,21,22,23]. Male menopause or andropause refers to a generalized decline in male hormones, including testosterone and dehydroepiandrosterone, in middle-aged and elderly men. This decrease has been associated with a number of changes, including depression, loss of libido, sexual dysfunction, and changes in body composition [24]. Additionally, the decline in testosterone with age has been associated with specific physical changes that affect quality of life and life expectancy [25].

Enoki mushroom extract and its compounds as well as their potential health-promoting effects have been examined using in vitro and in vivo animal experiments [7,8,9,10]. Extensive exploratory research on natural food materials that increase testosterone secretion as a countermeasure for male menopause recently identified enoki mushroom as a food that promotes testosterone secretion [11]. In the present study on the total score and subscales of AMS in healthy men with AMS scores of 27–49 at baseline, the intake of enoki mushroom extract (the test food) for 12 weeks significantly improved the sexual subscale of AMS in a stratified analysis of subjects with low testosterone levels (Table 3). Furthermore, in a careful examination of the changes after 12 weeks on the test food in the AMS scores of subjects with severe symptoms who scored 4 or 5 in each question of the questionnaire at baseline, the cumulative χ^2^ test revealed that the AMS scores were significantly lower in the test food group than in the placebo group (Table 5).

Iguchi et al. [11] reported that the administration of enoki mushroom extract increased testosterone production in a cisplatin-impaired mouse model. Additionally, the administration of this extract to wet-floor-fatigue-model mice promoted testicular testosterone production and enhanced Leydig cell function. In this study, however, there were no significant differences between the test and placebo groups in both total and free testosterone levels (Table 6). Stratified analysis of subjects with borderline LOH (8.5 pg/mL ≤ free testosterone < 11.8 pg/mL) also showed the same result. On the other hand, a stratified analysis of subjects divided into two groups based on a change in total testosterone of <0.5 ng/mL and >0.5 ng/mL after 12 weeks of intake showed that the number of subjects with increases in total testosterone of ≥0.5 ng/mL was significantly higher in the test food group than in the placebo group (Table 7). The borderline value for total testosterone in LOH syndrome is between 2.5 and 3.0 ng/mL, so this increase of 0.5 ng/mL of total testosterone may enable escape from the borderline state. In this sense, this stratification analysis seems meaningful. The results suggest that the enoki mushroom extract may work to promote testosterone secretion in some people.

Adenosine, a component of enoki mushroom extract, is known to increase testosterone production. Iguchi et al. [11] examined the effects of enoki mushroom extract on testosterone production both in vivo and in vitro, and confirmed that the main pharmacologically active ingredient present in enoki mushroom extract was adenosine, a ubiquitous biological component. Measurements of adenosine concentrations and testosterone secretion-promoting activity showed that adenosine enhanced testosterone secretion-promoting activity in a concentration-dependent manner. Additionally, the administration of enoki mushroom extract increased testosterone production in the fatigue model, and adenosine supplementation exerted similar effects [11]. Therefore, adenosine may have been partially responsible for the attenuation of male menopausal symptoms in the present study as the pharmacologically active constituent of enoki mushroom extract.

In men, testosterone is mainly synthesized in Leydig cells in the testes and is then secreted throughout the body. Therefore, male menopausal symptoms and LOH syndrome may be attenuated by promoting testosterone secretion by Leydig cells. Enoki mushroom extract effectively increased testosterone production and contributed to the restoration of testicular function in animal models of fatigue and aging [11]. Collectively, the present results indicate that enoki mushroom extract containing high adenosine levels may be useful against fatigue and aging in humans. Furthermore, the testosterone production-increasing effect of enoki mushroom extract may effectively promote recovery from stress and stress-induced fatigue.

In conclusion, the present study demonstrated for the first time that enoki mushroom extract may exert beneficial effects on male menopausal symptoms and LOH in middle-aged and elderly men.

## Figures and Tables

**Figure 1 nutrients-17-01208-f001:**
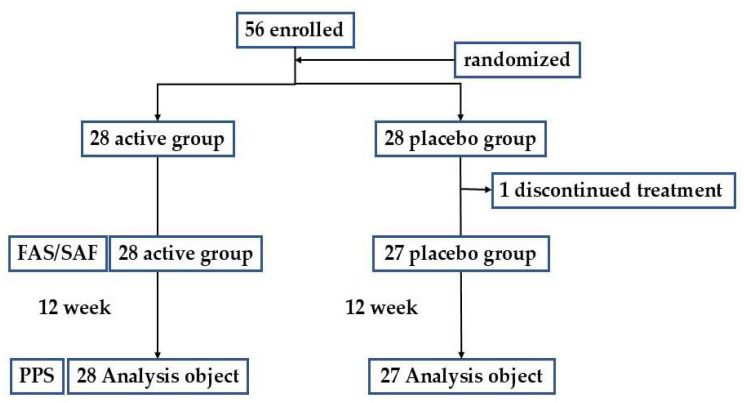
Flow diagram of participants throughout the study. FAS: Full Analysis Set; SAF: Safety Analysis Set; PPS: Per Protocol Set.

**Table 1 nutrients-17-01208-t001:** Baseline characteristics of subjects in test food (enoki mushroom extract) and placebo groups.

Variable	Test Food Group (n = 28)		Placebo Group (n = 27)		*p* Value
	Mean	SD	Mean	SD	
Age (years)	62.00	11.25	61.48	10.85	0.863
Height (cm)	169.00	6.15	169.58	5.51	0.714
Weight (kg)	68.58	12.51	69.15	10.84	0.859
BMI (kg/m^2^)	23.88	3.40	24.03	3.48	0.879
Blood pressure (mmHg)					
Systolic	130.50	18.73	131.26	14.40	0.867
Diastolic	78.04	12.38	77.67	9.97	0.904
Pulse	74.04	10.73	74.74	14.38	0.837
Male Menopausal Symptom Rating Scale					
AMS	35.86	7.77	34.52	7.15	0.510
BDI-Ⅱ	9.25	5.90	6.26	4.52	0.040
N-QOL	84.09	13.72	90.29	11.60	0.077
PSQI	5.75	2.91	5.26	2.93	0.536
ASE	28.29	4.54	28.96	4.03	0.561
Total testosterone	5.08	2.00	5.40	2.33	0.587
Free testosterone	9.19	3.39	9.64	3.09	0.609
Blood Biochemistry					
AST(GOT) (U/L)	20.68	4.85	24.04	14.49	0.251
ALT(GPT) (U/L)	19.61	7.90	25.33	21.03	0.193
γ-GTP (IU/L)	36.57	28.15	55.11	46.73	0.083
Total cholesterol (mg/dL)	207.82	34.48	212.41	30.24	0.603
Triglycerides (mg/dL)	138.93	95.26	189.96	133.81	0.108
HDL cholesterol (mg/dL)	59.57	13.64	53.52	11.19	0.078
LDL cholesterol (mg/dL)	118.96	29.05	122.52	33.10	0.674
HbA1c (NGSP) (%)	5.58	0.38	5.47	0.37	0.307
eGFR (mL/min/1.73 m^2^)	65.05	10.58	67.51	13.25	0.451

Differences between the placebo and test food groups were analyzed using Student’s *t*-test for means.

**Table 2 nutrients-17-01208-t002:** Comparison of AMS scores in test food (enoki mushroom extract) and placebo groups by analysis of covariance (ANCOVA).

Variable		Test Food Group (n = 28)		Placebo Group (n = 27)		Source	*SS*	*df*	*MS*	*F*-Value	*p* Value
		Mean	SD	Mean	SD						
Total score	Pre	35.86	7.77	34.52	7.15	pre Total score	1178.32	1	1178.32	25.38	0.000
	Post	27.61	8.07	26.78	6.38	Interaction	1.53	1	1.53	0.03	0.857
						Error	2414.11	52	46.43		
						Overall	3596	54			
Psychological subscale	Pre	15.25	4.48	15.07	3.22	pre Psychological subscale	341.47	1	341.47	27.94	0.000
	Post	11.61	4.24	10.96	3.11	Interaction	4.66	1	4.66	0.38	0.540
						Error	635.62	52	12.22		
						Overall	980.11	54			
Somatic subscale	Pre	7.79	2.59	7.37	2.37	pre Somatic subscale	157.16	1	157.16	41.29	0.000
	Post	6.43	2.33	5.93	1.77	Interaction	1.92	1	1.92	0.5	0.481
						Error	197.94	52	3.81		
						Overall	355.2	54			
Sexual subscale	Pre	12.82	2.96	12.07	3.3	pre Sexual subscale	170.65	1	170.65	18.16	0.000
	Post	9.57	3.65	9.89	2.94	Interaction	5.49	1	5.49	0.58	0.448
						Error	488.67	52	9.4		
						Overall	674.91	54			

Data were analyzed via analysis of covariance (ANCOVA). *SS* = sum of squares; *df* = degrees of freedom; *MS* = mean square.

**Table 3 nutrients-17-01208-t003:** Comparison of AMS scores in test food (enoki mushroom extract) and placebo groups with low testosterone levels (borderline cases of LOH syndrome) via analysis of covariance (ANCOVA).

Variable		Test Food Group (n = 11)		Placebo Group (n = 13)		Source	*SS*	*df*	*MS*	*F*-Value	*p* Value
		Mean	SD	Mean	SD						
Total score	Pre	35.73	7.67	36.08	8.15	pre Total score	315.94	1	315.94	7.82	0.011
	Post	25.27	6.42	27.23	8.24	Interaction	18.77	1	18.77	0.46	0.503
						Error	848.48	21	40.4		
						Overall	1179.83	23			
Psycholo-gical subscale	Pre	16.18	5.06	15.38	3.4	pre Psychological subscale	109.42	1	109.42	6.7	0.017
	Post	11.36	5.08	11.08	3.8	Interaction	0.05	1	0.05	0	0.957
						Error	342.98	21	16.33		
						Overall	453.96	23			
Somatic subscale	Pre	6.73	2.2	8.31	2.84	pre Somatic subscale	51.07	1	51.07	17.09	0.000
	Post	5.82	1.4	6.38	2.36	Interaction	0.03	1	0.03	0.01	0.920
						Error	62.77	21	2.99		
						Overall	119.96	23			
Sexual subscale	Pre	12.82	2.75	12.38	3.23	Pre Sexual subscale	52.22	1	52.22	11.07	0.003
	Post	8.09	1.7	9.77	3.14	Interaction	21.18	1	21.18	4.49	0.046 *
						Error	99.04	21	4.72		
						Overall	177.83	23			

Data were analyzed via analysis of covariance (ANCOVA). *SS* = sum of squares; *df* = degrees of freedom; *MS* = mean square, * = *p* < 0.05.

**Table 4 nutrients-17-01208-t004:** Number of subjects in test food (enoki mushroom extract) and placebo groups showing changes in AMS scores after 12 weeks of intake.

Food	-	0	+1	+2	+3	+4	Total
Test food group	1	4	13	9	13	2	42
Placebo group	1	8	16	8	4	0	37
Total	2	12	29	17	17	2	79

Score changes were expressed as follows: -, decreased; 0, unchanged; +1 to +4, increased.

**Table 5 nutrients-17-01208-t005:** Chi-analysis of cumulative number of subjects in test food (enoki mushroom extract) and placebo groups based on degree of improvement in AMS scores.

	X1^2^		X2^2^		X3^2^		X4^2^		X5^2^	
Food	-	0~+4	-~0	+1~4	-~+1	+2~4	-~+2	+3~4	-~+3	+4
Test food group	1	41	5	37	18	24	27	15	40	2
Placebo group	1	36	9	28	25	12	33	4	37	0
Total	2	77	14	65	43	36	60	19	77	2

χ^2^ adj = 11.35, *df* = 3.68, *p* = 0.018.

**Table 6 nutrients-17-01208-t006:** Mean and standard deviation (SD) of total and free testosterone levels in test food and placebo groups.

Total or Free Testosterone	Test Food Group (n = 28)		Placebo Group (n = 27)		Total (n = 55)	
	Mean	SD	Mean	SD	Mean	SD
Total testosterone (ng/mL)	0.63	1.10	0.37	1.34	0.50	1.22
Free testosterone (pg/mL)	0.05	1.71	−0.33	1.62	−0.13	1.67

**Table 7 nutrients-17-01208-t007:** Number of subjects in test food and placebo groups, divided as change in total testosterone of <0.5 ng/mL and of >0.5 ng/mL.

Total Testosterone	−2.4~0.49 ng/mL	0.5~4.9 ng/mL
Test food	11	17
Placebo	18	9

The numbers of subjects in the test food and placebo groups were analyzed using the cumulative χ^2^ test (χ^2^ = 4.134, *df* = 1, *p* = 0.042).

## Data Availability

The original contributions presented in this study are included in the article. Further inquiries can be directed to the corresponding author.

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
