# Peer review of "Beneficial Effects of Enoki Mushroom Extract on Male Menopausal Symptoms in Japanese Subjects: A Randomized, Double-Blind, Placebo-Controlled Study"

_nutrients, 2025, doi:10.3390/nu17071208_

Round 1

Reviewer 1 Report

Comments and Suggestions for Authors

The present study is really interesting and of great importance, as it highlights significant problem in some men – male menopause. Please find the following remarks that I believe, will improve the quality of the manuscript.

Introduction

-Rows 38-39 – „Among them, testosterone is related to sexual dysfunction as well as muscle strength, cognitive decline, higher mental functions, and other physiological functions.” – is it really what you want to say? Is testosterone related to sexual dysfunction? I hope you meant the decline in testosterone level. Please reformulate this sentence.

-Rows 42-43 – can you elaborate on it and briefly describe what kind of diet should be recommended to prevent the age-related decline in testosterone levels?

-The same remark refers to the paragraph concerning enoki mushrooms. It would be valuable to briefly present their nutritional value and the most significant bioactive compounds responsible for the observed health benefits of enoki mushrooms.

-Row 50 – a space is needed

-You describe one study with an animal model in which enoki mushrooms extract was administered. Are there any studies involving human subjects using enoki extract as well?

Materials and Methods

-How did you choose the exact amount of the powdered enoki mushrooms extract (125 mg in each capsule)? Is it based on the results of any studies indicating that such an amount may be sufficient to observe any positive health effects? The same question is asked for the total daily amount of the powdered extract (10 capsules x 125 mg of extract). You only explain that the safety of such an administration was confirmed, but what about its effectiveness?

-Rows 92-93 – what was the rationale for setting one of the inclusion criteria AMS score of 27-49? Moreover, in the sub-paragraph concerning the AMS scale, it is necessary to present cut-off points – which scores may be interpreted as the right one and which as the abnormal one? What is the maximum score to obtain? And what does it indicate? (less intensity of menopausal symptoms?)

-Row 114 – as you use a lot of abbreviations indicating blood parameters, it would be better to explain them and place after “Conflict of interest” information (at the end of the manuscript).

-Figure 1 – all the abbreviations must be explained.

-Table 1 – as I can see, different male menopausal symptom rating scales were used, while only AMS was described. Either you delete the results of the rest of the scales or you will described all the scales in the Materials and Methods section.

Author Response

Responses to the comments by Reviewer # 1

We thank the reviewer for his constructive comments for our manuscript (MS), in order to improve the quality of the manuscript. We agree with his positive and valuable comments and concerns, and thus we revised the MS according to the comments, with our point-to-point replies. All revisions in the text and Tables of revised MS were marked in red color.

Comments and Suggestions for Authors

The present study is really interesting and of great importance, as it highlights significant problem in some men – male menopause. Please find the following remarks that I believe, will improve the quality of the manuscript. 

Question, Comments

Introduction

-Rows 38-39 – „Among them, testosterone is related to sexual dysfunction as well as muscle strength, cognitive decline, higher mental functions, and other physiological functions.” – is it really what you want to say? Is testosterone related to sexual dysfunction? I hope you meant the decline in testosterone level. Please reformulate this sentence.

Answer

Thank you for a convincing comment. I agree with the meaning of your indication. Based on the suggestion, we revised the sentence to clarify the meaning, as follows (P 1):

 “Among them, the decline of testosterone may be related to sexual dysfunction as well as loss of muscle mass, cognitive decline, and other physiological dysfunction.”

Question, Comments

-Rows 42-43 – can you elaborate on it and briefly describe what kind of diet should be recommended to prevent the age-related decline in testosterone levels?

Answer

Thank you for a reasonable comment. We could not elaborate exactly about the precise diet to prevent the age-related decline in testosterone, based on our results. However, generally, it is recommended in the literature, that protein-diet is considered a key nutrient for aging population. The most information on dietary protein requirement for older adults derives from studies in healthy “disease-free” older individuals. Also, the observational studies in older people suggests a relation between lower dietary protein intake and loss of muscle mass. Certain diets such as Mediterranean diet, may play a significant role in healthy aging by preventing the onset of certain disease and by improving the aging process itself, such as male menopausal symptoms. Furthermore, natural polyphenolic compounds (antioxidants) may be able to improve the activities of steroidogenic enzymes and testosterone bioavailability which cause the prevention of male late-onset hypogonadism. Overall, the well-rounded diets full of healthy fats, proteins, and nutrient-dense fruits and vegetables may support healthy testosterone production.

Therefore, we described briefly about the possible diet to prevent the age-related decline in testosterone levels, in the revised MS, as follows (P 2):

“ Generally, potein-diet, with the natural polyphenolic antioxidants, is considered a key nutrient for aging population.

Question, Comments

-The same remark refers to the paragraph concerning enoki mushrooms. It would be valuable to briefly present their nutritional value and the most significant bioactive compounds responsible for the observed health benefits of enoki mushrooms.

Answer

We agree with the comment by the reviewer that the same remark refers to the enoki mushrooms paragraph. So, we described more briefly the nutritional values and the bioactive compound (adenosine) for the health benefits of enoki mushrooms by deleting two duplicated sentences in the old MS, as follows (P 2, lines 49-60):

 Enoki mushroom (Flammulina velutipes), one of the main edible mushrooms on the planet, has long been recognized for its nutritional value and delicious taste [7]. Previous studies on enoki mushroom showed that it possessed various biological and pharmacological properties, including anticancer, antimicrobial, antioxidant, and immunomodulatory activities [7-10]. Iguchi et al. [11] recently reported that the administration of an ethanolic extract of enoki mushroom increased testosterone production in a cisplatin-impaired model of mice, and suggested the involvement of adenosine as the most significantly active component. The administration of enoki mushroom extract or adenosine itself to wet floor fatigue model mice was also shown to promote testicular testosterone production and enhanced Leydig cell function. Based on these findings, they has concluded that enoki mushroom containing a high content of adenosine may be useful against aging and fatigue.

Question, Comments

-Row 50 – a space is needed

Answer

  Thank you for a kind indication of our mistake. We inserted a space.

【Question, Comments】

-You describe one study with an animal model in which enoki mushrooms extract was administered. Are there any studies involving human subjects using enoki extract as well?

Answer

To our knowledge, this is a first human intervention study for the male menopausal symptoms, using our enoki extract.

Question, Comments

Materials and Methods

-How did you choose the exact amount of the powdered enoki mushrooms extract (125 mg in each capsule)? Is it based on the results of any studies indicating that such an amount may be sufficient to observe any positive health effects? The same question is asked for the total daily amount of the powdered extract (10 capsules x 125 mg of extract). You only explain that the safety of such an administration was confirmed, but what about its effectiveness?

Answer

Thank you for a reasonable and critical question about the amount and daily dose of enoki mushroom extract for the human intervention study. Basically, the possible total daily dose (1250 mg) for the effectiveness in human by the powdered enoki mushroom extract was predicted by the estimation of adenosine (significant active ingredient) content on the basis of our previous preclinical (animal) experiments (Iguchi et al., Ref 11). In fact, to determine the effective dose of the enoki extract in the present study, we estimated the daily total dose (1250 mg) of powered enoki mushrooms by converting per body weight in human (generally 60 kg) and rats by using the pharmacologically effective dose in rats. The dose (125 mg) of enoki extract was included per capsule because of relatively large amount of 1250 mg extract, and 10 capsules per day were ingested for each subject in the present study. Therefore, to clarify this issue, we stated in the text of the revised MS, as follows (P 2):

“The daily oral dose (1250 mg) of enoki mushroom extract was determined by converting per body weight in human and rat on the basis of adenosine content in previous preclinical study (11).”

Question, Comments

-Rows 92-93 – what was the rationale for setting one of the inclusion criteria AMS score of 27-49? Moreover, in the sub-paragraph concerning the AMS scale, it is necessary to present cut-off points – which scores may be interpreted as the right one and which as the abnormal one? What is the maximum score to obtain? And what does it indicate? (less intensity of menopausal symptoms?)

Answer

Thank for a critical comment of the inclusion criteria AMS score in the study. In general, AMS score of 50 or more is defined as the requirement of medical treatment for the definition of patient. The target persons in this study were healthy people (borderline person, not patient) with menopausal symptoms. According the general definition, we defined AMS scores as 17-26 points (None), 27-36 points (Mild), 37-49 points (Moderate), and 50 points or more (Severe). To clarify this issue, we added the following sentence in the revised MS, as follows (P 2):

“……with AMS scores of 27-49, who had mild or moderate menopausal symptoms (Definition of Late Onset Hypogonadism by “The Japanese Urological Association”, 2007).”

Question, Comments

-Row 114 – as you use a lot of abbreviations indicating blood parameters, it would be better to explain them and place after “Conflict of interest” information (at the end of the manuscript).

Answer

  Thank you for a reasonable indication of abbreviations in the text. As pointed out by the reviewer, we explained clearly all abbreviations in the text (P 4) and also in the last place after “Conflict of interest” (at the end of manuscript and before the literature list) (P 9)

Abbreviations: AMS: Heinemann Aging Males’ Symptoms score, PSA: prostate-specific antigen, BMI: body mass index, BDI-II: Beck Depression Inventory, N-QOL: Nocturia Quality-of-Life Questionnaire, PSQI: Pittsburgh Sleep Quality Index, ASE: activity scale for the elderly, Hb: hemoglobin, Ht: hematocrit, AST: aspartate aminotransferase, ALT: alanine aminotransferase, g-GTP: gamma-glutamyl transpeptidase, ALP: alkaline phosphatase, LDH: lactate dehydrogenase, CPK: creatine phosphokinase, HDL: high density lipoprotein, LDL: low density lipoprotein, eGFR: estimated glomerular filtration rate, FAS: full analysis set, SAF: safety analysis set, PPS: per protocol set.

Question, Comments

-Figure 1 – all the abbreviations must be explained.

Answer

  We explained the abbreviations in Fig 1, as pointed out by the reviewer.

Question, Comments

-Table 1 – as I can see, different male menopausal symptom rating scales were used, while only AMS was described. Either you delete the results of the rest of the scales or you will described all the scales in the Materials and Methods section.

Answer

 Thank you for a reasonable comment. We described about the measurements of all the scales in the “Materials and Methods” measured as secondary endpoints.

Reviewer 2 Report

Comments and Suggestions for Authors

The study reports the results of a randomized double-blind study on two groups of elderly people, one of which was treated with enoki mushroom capsules, to evaluate the evolution of male menopausal symptoms.

The experimental design is correct. The inclusion and exclusion criteria are fully explained. The details provided about the experiment allow its replication.

The authors correctly interpret the results of the study without excessive emphasis, declaring that "enoki mushroom extract containing high adenosine levels may be useful against fatigue and aging in humans". The experiment is interesting and deserves to be published.

Author Response

Responses to the comments by Reviewer # 2

  We thank the reviewer for his thoughtful comments for our manuscript (MS). We agree with his valuable and reasonable comments and concerns, and thus we revised the MS according to the comments, with our point-to-point replies. All revisions in the text and Tables of revised MS were marked in red color.

Question, Comments

Comments and Suggestions for Authors

The study reports the results of a randomized double-blind study on two groups of elderly people, one of which was treated with enoki mushroom capsules, to evaluate the evolution of male menopausal symptoms. The experimental design is correct. The inclusion and exclusion criteria are fully explained. The details provided about the experiment allow its replication. The authors correctly interpret the results of the study without excessive emphasis, declaring that "enoki mushroom extract containing high adenosine levels may be useful against fatigue and aging in humans". The experiment is interesting and deserves to be published.

Answer

  Thank you for a positive and encouraging comment.

Reviewer 3 Report

Comments and Suggestions for Authors

The experiment planned properly and conducted methodically.

The results processed properly.

The presentation of the results is appropriate, the discussion is appropriate.

I suggest only to move the Supplementary Table from Supplementary material to the main article.

Author Response

Responses to the comments by Reviewer # 3

  We thank the reviewer for his thoughtful comments for our manuscript (MS). We agree with his valuable and reasonable comments and concerns, and thus we revised the MS according to the comments.

Comments and Suggestions for Authors

【Question, Comments】

The experiment planned properly and conducted methodically.

The results processed properly. The presentation of the results is appropriate, the discussion is appropriate.

I suggest only to move the Supplementary Table from Supplementary material to the main article.

【Answer】

  Thank you for a positive and reasonable comment. As suggested by the reviewer, we moved the Supplementary Table (as Table 7) in the text (P 6, line 224) from the Supplementary material (P 8).